# Acrylamide in Bakery Products: A Review on Health Risks, Legal Regulations and Strategies to Reduce Its Formation

**DOI:** 10.3390/ijerph18084332

**Published:** 2021-04-19

**Authors:** Cristina Sarion, Georgiana Gabriela Codină, Adriana Dabija

**Affiliations:** Faculty of Food Engineering, Stefan cel Mare University of Suceava, 720229 Suceava, Romania; sarioncristina@yahoo.com (C.S.); adriana.dabija@fia.usv.ro (A.D.)

**Keywords:** acrylamide, toxicity, public health, legal regulations, reduction method, bakery products

## Abstract

Acrylamide is a contaminant as defined in Council Regulation (EEC) No 315/93 and as such, it is considered a chemical hazard in the food chain. The toxicity of acrylamide has been acknowledged since 2002, among its toxicological effects on humans being neurotoxicity, genotoxicity, carcinogenicity, and reproductive toxicity. Acrylamide has been classified as carcinogenic in the 2A group, with human exposure leading to progressive degeneration of the peripheral and central nervous systems characterized by cognitive and motor abnormalities. Bakery products (bread, crispbread, cakes, batter, breakfast cereals, biscuits, pies, etc.) are some of the major sources of dietary acrylamide. The review focuses on the levels of acrylamide in foods products, in particular bakery ones, and the risk that resulting dietary intake of acrylamide has on human health. The evolving legislative situation regarding the acrylamide content from foodstuffs, especially bakery ones, in the European Union is discussed underlining different measures that food producers must take in order to comply with the current regulations regarding the acrylamide levels in their products. Different approaches to reduce the acrylamide level in bakery products such as the use of asparginase, calcium salts, antioxidants, acids and their salts, etc., are described in detail.

## 1. Introduction

Bakery products are considered staple foods in many countries due to their content of essential nutrients such as proteins, carbohydrates, fiber and vitamins [1]. In addition to these nutrients, bakery products may contain a number of compounds that are formed in them during heat treatment, such as acrylamide, hydroxymethylfurfural and their derivatives [2,3,4]. Assessing the presence and reducing the level of acrylamide formed in heat-treated foods is a major concern in many countries [5].

Acrylamide (AA) was first synthesized in 1949, and a year later it was used as a flocculating and thickening agent in the synthetic materials industry, for drinking water treatment, in cosmetics, in the pulp and paper industry, in the textile industry, in synthesis, dyes and gels, etc.

Acrylamide or acrylic acid amide is a chemical contaminant that is formed during the technological process of baking, frying or grilling certain foods at temperatures above 120 °C and in low humidity conditions [6,7].

Acrylamide is formed mainly in carbohydrate-rich foods, during the Maillard reaction between reducing carbohydrates (glucose, fructose, etc.) and amino acids (especially asparagine), a reaction responsible for the formation of specific taste and color (browning/ frying) [8,9,10].

Acrylamide is considered to have appeared since the discovery of fire and food cooking by methods of baking, frying and grilling, but then its toxic effects in humans and animals, namely nervous system damage, pre- and post-natal development, negative effects on the male reproductive system, the possibility of cancer and genotoxicity were not known [11,12,13,14,15,16].

The harmful effects of acrylamide on human health were discovered in 2002 by a group of Swedish researchers at the University of Stockholm, together with specialists from the Swedish National Food Administration, who sounded the alarm after finding that the population, through food, ingests a much higher amount of AA than the maximum limit allowed at that time in drinking water [17,18,19].

Nowadays, many researchers representing food safety authorities, academia, and food manufacturers have sought to better understand the kinetics and mechanisms of acrylamide formation, studied the influencing factors, asked themselves questions related to bioavailability and toxicity and finally, are trying to find in a continuous way solutions to minimize its formation in foods. Although nowadays a lot of information has become available, there are still important problems to be solution [1,20].

Factors that influence the content of acrylamide in heat-processed foods such as bakery ones are initial concentration of the precursors, their ratio, flour quality such as flour milling intensity, fermentation conditions, the thermal processing methods (for example, baking, frying, toasting), the processing conditions, such as the temperature, heating time, pH, water content and activity, physical state of the food, additives, etc. [1,4,5,21,22,23,24,25].

In recent years the food industry has proposed mitigation strategies toward reducing levels of acrylamide in its products while maintaining the quality parameters unaffected by the adjusted processing conditions [25,26]. These include modifying the product formulations or processing conditions (lowering pH, baking temperature and time), and adding food ingredients that have been reported to inhibit acrylamide formation (organic acids, mainly citric, calcium or magnesium ions, and extracts with antioxidant properties) [2,13]. Researchers have also found other solutions to lower acrylamide levels in foods: to breed wheat genotypes that have low levels of free asparagine concentration [27], such as using the enzyme asparaginase to hydrolyze asparagine to aspartic acid and ammonia prior to cooking or processing (to reduce acrylamide levels by 70–90% without affecting the organoleptic properties of the products) [25,28,29,30]; application of modern processes during the baking process such as radio frequency, inert atmosphere, steam baking, microwaves baking, etc. [4,31,32]; applying glycine and glutamine to dough prior to fermentation [28,33,34]; the use of bacteriocin like inhibitory substances producing lactic acid bacteria (LAB) with high proteolytic activity to perform the fermentation processes with yeast [1,29]; replacing reducing sugars with sucrose [34]; using the different additives, such as rosemary, amino acids or proteins [23,35] etc.

The purpose of this paper was to provide an overview of the toxicological effects of AA on human health, the legislation in force governing reference levels as shown in Figure 1 and methods of reducing AA levels in bakery products.

## 2. Toxicological Effects of Acrylamide on the Human Body and Its Risks Represented by It Consumption

Research has shown that AA is absorbed by humans and animals through ingestion, inhalation and skin. After inhalation AA is rapidly distributed to all organs of the body through the bloodstream. AA can be identified in the brain, heart, liver, kidneys and breast milk [36,37].

In the body, AA is metabolized to a chemical, reactive epoxide, glycidamide, following the reaction catalyzed by the cytochrome enzyme [38]. AA conversion to the reactive, mutagenic and genotoxic compound, glycidamide, can occur in both rodents and humans [39]. Glycidamide formation is considered responsible for the genotoxic effects of AA having the potential to induce mutagenic genes at the chromosomal level [40,41].

AA and glycidamide can react with macromolecules, such as hemoglobin, DNA, serum albumin and enzymes, to form adducts. The formation of adducts with DNA is likely to lead to the toxic, carcinogenic potential of AA [42].

Research on laboratory animals has shown that exposure to AA have the following effects: genotoxic, carcinogenic, neurotoxic, affects the male reproductive system and has effects on pre- and postnatal development. AA has contributed to genetic mutations and tumors in various organs. Based on animal experiments, the European Food Safety Authority (EFSA) experts have concluded that AA in food may increase the risk of cancer in consumers of all ages, including children, which is the most exposed age group [43,44,45].

The possible carcinogenic effects of AA have been discovered since 1994 when the International Agency for Research on Cancer (IARC) classified AA as having "carcinogenic potential in humans" (group 2A) [46,47,48].

In April 2002, the Swedish National Food Administration (SNFA) reported that AA is also found in food and that by eating common foods such as bread, biscuits, chips, coffee, French fries, people ingest AA in unknown amounts that can have negative effects on human health [19,49,50,51,52,53].

Later in the same year, 2002, the Food and Agriculture Organization of the United Nations/World Health Organization (FAO/WHO) organized a consultation to gather the views of an international group of experts on the health implications of AA in food. The aim was to review and evaluate new and existing AA data and research, identify needs for further information and studies and also to develop and suggest possible interim advice for governments, industry and consumers [45,54].

In 2005, the Joint FAO/WHO Expert Committee on Food Additives (JECFA, 2005) conducted an evaluation of available data on AA. The full report was published in 2006. Food intakes were estimated to be between 1 µg/kg (medium exposure dose) and 4 µg/kg (high exposure dose) in body weight/day for the general population and consumers with a high intake. These estimates also included children [46,55].

JECFA concluded that epidemiological studies and data on biomarkers available in humans and animals at the time of assessment were not adequate to establish a dose-response relationship and therefore performed the assessment based on available animal studies. JECFA considered the genotoxicity and carcinogenicity of AA to be essential effects for risk assessment, but also considered other non-cancerous endpoints of concern, such as effects on the nervous system. JECFA concluded that, at an estimated average consumption, morphological changes of the nerves could not be excluded for some individuals [45,55].

JECFA recommended continuing efforts to reduce AA concentrations in food and that AA be revaluated when the results of long-term carcinogenicity and neurotoxicity studies become available [45,56].

In April 2005, the EFSA Scientific Panel on Contaminants in the Food Chain (CONTAM Group) agreed with the main conclusions and recommendations of JECFA on AA risk assessment. The CONTAM Group noted the use of data from European countries, including information collected in collaboration initiatives between the European Commission and EFSA, and concluded that further evaluation by EFSA was not necessary at that time [31,45].

In 2005, the Panel of Experts of the National Centre for Toxicology for the Risk Assessment of Human Reproduction (NTP-CERHR) published a report on the toxicity of AA reproduction and development [57]. The report concluded that there are no human data available on the developmental or reproductive toxicity of AA and that the available experimental data were sufficient to conclude that AA may cause developmental toxicity in rats and mice. The NTP-CERHR expert group also concluded that there is sufficient data to conclude that AA induces transmissible genetic lesions in male germ cells of mice in the form of mutual translocations and genetic mutations.

In 2008, EFSA organized a scientific colloquium on AA carcinogenicity and new evidence related to dietary exposure [58,59,60]. The aim of this EFSA colloquium was to discuss the challenges regarding the potential toxicity and cancer risk associated with food exposure to AA, given the new information that has become available since the last risk assessment carried out by JECFA in 2005 [55].

In 2011, JECFA the researchers concluded that the average dose of exposure to AA is 0.2–1.0 µg/kg body weight/day for the adult population, while 95% of cases were in the range of 0.6–1.8 µg/kg body weight/day [61].

In 2015, the Scientific Group for Food Chain Contaminants (CONTAM) of the European Food Safety Authority adopted an opinion on AA in food. Based on animal studies, the Authority confirms previous assessments that food acrylamide may increase the risk of developing cancer for consumers in all age groups [62]. Given that acrylamide is present in a wide range of foods consumed daily, this warning applies to all consumers, but children are the most exposed age group based on body weight. The possible harmful effects of acrylamide on the nervous system, on prenatal and postnatal development and on male reproduction were not considered to be a cause for concern, based on current levels of dietary exposure. Current levels of dietary exposure to acrylamide in all age groups indicate a concern about its carcinogenic effects [32,45,63,64].

## 3. Legislative Rules on the Maximum Benchmark Levels of Acrylamide in Bakery Products

Depending on the culinary habits of various countries, the intake of AA in the daily diet can vary between 10–30% for bakery products and 10–20% for pastries. Thus, in Germany, bread and rolls represent approximate 25% of the daily intake of AA, whereas in the Netherlands and Belgium, bread represents 10% of the daily intake of AA, and in Sweden up to 17%. In Romania, bread contributes between 14–37.5% of the maximum allowed dose of AA for food in the daily diet, estimated by FAO/WHO [60].

Thus we can consider that among the foods with the highest intake of AA in the daily diet are bread and bakery products [65]. Some researchers have reported that the highest level of AA is found in the crust of bread and less in bread crumbs [66].

Since 2005 EFSA, through the Scientific Committee for Contaminants in the Food Chain, recognizes the presence of AA in food and together with food industry operators, specialists and researchers from member countries have researched AA training pathways and developed a set of measures to reduce the level of this contaminant in food.

Thus, in 2007, the European Commission issued 2007/331/EC, on monitoring the level of AA in food which provides for a three-year monitoring program (2007–2009) of AA in certain foods, in order to provide concrete information on foodstuffs that have a high level of acrylamide or those that by consumption high food have a significant contribution to its assimilation. The categories of bakery products covered by the monitoring program are: bread; breakfast cereals; biscuits, including infant biscuits; baby food preserved in glass containers which may or not contain cereals; processed baby food based on cereals; other products (containing cereal products and baby food, other than those mentioned above).

The Recommendation of the European Commission (EC) 2007/331/EC [67] requires that sampling by member states to be carried out in both production and marketing units (e.g., supermarkets, smaller shops, bakeries, restaurants, etc.). A minimum number of samples were required for each product category, for each country, and a unitary model for reporting the types of food taken and the results obtained. A total number of 2042 samples/year were established, and Romania planned a number of 80 samples, with eight annual samples for each category of food product, respectively. During the 2017–2019 periods in Romania the National Sanitary Veterinary and Food Safety Authority have been analyzed a total of 138 samples from which 50 were analyzed in 2018 and 88 in 2019. No foods samples have been analyzed in 2017. However, in a complementary way the National Research & Development Institute for Food Bioresources, IBA Bucharest have been analyzed AA level from various foods products from Romania market. A number of 55 cereal products were evaluated in Romania during 2017–2018 periods [54,59].

Worldwide, in recent years many new methods have been developed for the determination of AA in food products, such as: high performance liquid chromatography mass spectrometry (HPLC–MS), high performance liquid chromatography tandem mass spectrometry (HPLC–MS/MS), gas chromatography-electron capture detection (GCECD), gas chromatography-mass spectrometry (GC-MS), liquid chromatography (LC) with ultraviolet (UV), or MS detection, and capillary electrophoresis (CE) with UV and MS, [22,24,56,68,69,70], solid-phase extraction (SPE) (were used to analyze acrylamide in bread), and dispersive liquid-liquid microextraction (DLLME) [71]. The ultra-performance liquid chromatography (UPLC) method has the advantage that the resolution is improved with higher analytical sensitivity and a shorter retention time [72]. The European Committee for Standardization published a method for AA detection in bakery products through liquid chromatography tandem mass spectrometry (LC-ESI-MS/MS) described in EN standard 16618:2015. In 2017 the European Committee for Standardization (CEN) published a technical specification for AA detection through gas chromatography (GC) coupled with mass spectrometry (MS) method in FprCEN/TS 17083 [73]. In Romania, at the national authority level, the laboratory accredited to perform AA determination uses high performance liquid chromatography coupled with a diode array detector (HPLC-UV). The separation of AA is done by passing on the chromatographic column followed by identification, following the series of photodiodes of the diode array detector (DAD) detector and identification by comparing TR (retention times) with TR of a standard substance. Also the AA confirmation is made by analyzing the spectrum obtained on the DAD detector. The interpretation of the results consists in identifying the compounds of interest by comparing the retention times of the peaks in the recorded chromatograms and by analyzing the DAD spectrum for the samples to be analyzed with the retention times of the peaks in the chromatograms recorded for the standard substance. The method present a good sensitivity: LOQ and LOD were 50 and 20 μg kg^−1^ for potato, coffee and potatoes products and 25 and 20 μg kg^−1^ for cereals, bread and bakery products respectively. All the methods used for AA determination are time-consuming and expensive. Therefore, nowadays specialists are trying to find rapid and low costs solutions for AA determinations. One of the quick methods for assessing the AA content from bakery products is to correlate some color values such as browning and HMF content with AA value. This method is only a predictive one and applicable only for heat products such as bakery ones [31].

The sampling method chosen by the Member States, in order to ensure the representativeness of the samples for the batch to be sampled, must follow the sampling procedures provided for in Regulation 2007/1234/EC. Also, the regulation 2007/333/EC is presenting also the methods of sampling and analysis for the official control of the levels of lead, cadmium, mercury, inorganic tin, 3-MCPD and benzo(a)pyrene in foodstuffs [74,75]. The reporting of the values obtained by AA is done taking into account the limit of quantification (LOQ), and to guarantee the comparability of the results, the analysis methods must be chosen so that the LOQ is 30 μg/kg for bread and bakery products. If, repeatedly, for the same product type, the AA value is below the LOQ, it will be replaced with another product [76].

In the case of bakery products, the specific information to be provided is: type of bread, soft or crunchy, fiber content, type of cereals, fermented or unfermented, type of fermentation (e.g., yeast), and other ingredients. The choice of the type of bread sampled must reflect the eating habits of each country [5].

In 2009, EFSA published in the scientific report entitled "*Results on the monitoring of acrylamide levels in food*" the results for 2007 and concluded that the measures to reduce the level of acrylamide were not constantly applied by food industry operators (OIA), and the toolbox developed by Food Drink Europe to support economic operators has not achieved the desired effects [77,78].

At the same time, it was found that an improvement of the monitoring program is necessary, regarding the food classification, and thus in 2010, the European Commission publishes the Recommendation 2010/307/EC on monitoring the level of acrylamide in food [79].

With this recommendation, ESFA considers it important to continue collecting data for annual monitoring of AA levels, and for food business operators and Member State authorities to continue research into ways to form acrylamide and how to reduce it.

The novelty of this recommendation is that in order to identify the evolution over time, it is important that products with the same specifications (e.g., the same type of bread) be sampled, if possible, every year. Specific information to be provided for bakery products: type of bread, e.g., wheat, rye, multigrain bread, bread with other ingredients, etc.

Following the two recommendations of the European Union, the Recommendation 2007/331/EC and the Recommendation 2010/307/EC which required the monitoring of acrylamide levels in certain foods during 2007–2010, the European Food Safety Authority (EFSA) published, in 2012, the results of monitoring in the scientific report entitled "Update on acrylamide levels in food from monitoring years 2007–2010” [77]. The report concluded that there was no steady trend in food groups towards lower acrylamide levels and that a decrease in acrylamide levels was observed only in certain food categories, while in other food categories, an increase in these levels was observed.

Thus, in 2013, the European Commission published the Recommendation 2013/647/EC, regarding the analysis of acrylamide levels in foods in which the indicative values for acrylamide are presented, based on EFSA monitoring data for the period 2007–2012 [80].

As AA levels are much higher in certain foods than in similar products in the same product category, the European Commission has recommended that surveys be carried out by the competent authorities of the member states to examine the production and processing methods used by OIA. These surveys will be conducted in case of exceeding the AA level above the indicative values established in this recommendation.

These investigations aim at analyzing the risks and critical control points, the HACCP system of the food business operator, to verify that the relevant technological steps in which acrylamide can form have been correctly identified and that appropriate measures have been taken to reduce them. The indicative values, set out in this Recommendation, do not constitute safety thresholds.

Table 1 shows the indicative values for acrylamide, based on EFSA monitoring data for the period 2007–2012, according to Recommendation 2013/647/EU (European Commission, 2013) and 2018–2021 period according to Recommendation 2017/2158/EU (European Commission, 2017) [80,81].

In 2017 the European Commission publishes Regulation 2017/2158/EC—establishing mitigation measures and reference levels to reduce the presence of acrylamide in food [81].

The regulation lays down measures to reduce the levels of acrylamide to be taken by food business operators, if the raw materials contain its precursors, so that these levels are below the reference levels set by the regulation.

The regulation applies to food business operators who produce and place on the market the following bakery products: bread and fine bakery products, breakfast cereals, baby food and food products (which may contain cereals), processed from cereals for small children.

Food business operators covered by this Regulation must identify, depending on the type and nature of their activities, the stages of food processing that are susceptible to the formation of acrylamide and determine, in the context of risk analysis and in accordance with the recommendations of this Regulation, appropriate measures to reduce acrylamide levels.

Diminution measures, in the case of producers who distribute products at county, national, or global level, should be included by the OIA in the procedures of the risk analysis system and critical control points (HACCP) of the production unit, or in the procedures good hygiene practices. The effectiveness of mitigation measures should be verified by sampling and analysis, for which analytical requirements and sampling frequency should be established, to ensure that the analytical results obtained are representative. This requirement does not apply to OIAs that produce food locally and perform only local retail activities. The values obtained for AA must be below the reference levels established by this Regulation. In order to verify the correct implementation of the measures to reduce the level of AA and to carry out the correct sampling and analysis, official controls will be performed by the authorities of the Member States [82].

Within this regulation, the reference levels have been revised, which are performance indicators, which aim to verify the correctness of the application of the reduction measures. These reference values shall be set at a very low level and shall be periodically reviewed every three years in accordance with this Regulation so as to ensure that the level of acrylamide is reduced to a very low level [82].

It is maintained the recommendation that the sampling and analysis of AA be performed according to Regulation 2007/333/EC (European Commission, 2007) presenting the methods of sampling and analysis for the official control of the levels of lead, cadmium, mercury, inorganic tin, 3-MCPD and benzo (a) pyrene in foodstuffs [75].

The reference levels shown in Table 1 for detecting the presence of acrylamide in bakery products, according to 2017/2158/EC (European Commission, 2017) laying down reduction measures and reference levels for reducing the presence of acrylamide in foodstuffs.

Because it was concluded that Regulation 2017/2158/EC did not present sufficient available data on the presence of acrylamide in foods [81] was adopted in 2019 by the European Commission the Recommendation 2019/1888/EC—on monitoring the presence of acrylamide in certain foods including bakery products [83]. This normative act adds a new list of non-exhaustive food products, which must be monitored in order to identify the risks and adopt new prevention and/or reduction measures against this contaminant.

The new food list adds the following specialties for bakery products: buns (hamburger buns, whole wheat buns and milk buns, etc.); sticks, Mexican tortillas; horns; doughnuts; bread specialties (e.g., pumpernickel bread, olive ciabatta, onion bread, etc.); pancakes; crispy cookies made of a thin layer of dough and fried in oil; churros.

The obligation to monitor AA levels and the effectiveness of measures to reduce it, as well as that of Member States and OIA to transmit to EFSA, the data collected in the previous year, in order to compile a database, remains.

The recommendation is maintained that the sampling and analysis of AA be performed according to Regulation 2007/333/EC (European Commission, 2017) laying down the methods of sampling and analysis for the official control of the levels of lead, cadmium, mercury, inorganic tin, 3-MCPD and benzo(a)pyrene in foodstuffs.

By adopting the Recommendation 2019/1888/EC (European Commission, 2019), the Recommendations 2010/307/EC (European Commission, 2010) and 2013/647/EC (European Commission, 2013) are repealed [79,83,84].

## 4. Methods to Reduce the Acrylamide Content in Bakery Products

In 2005 the Food Drink Europe Organization, who represents economic operators in the EU food and beverage industry, developed a set of “acrylamide toolbox” tools, which includes the most complete information from authorities, scientists, international research organizations and economic operators, about the ways of forming AA and the methods of reducing it from certain food groups. This “toolkit” is revised periodically, the 15th version being published in 2019. This material presents 14 different parameters (“tools”), grouped into four broad categories (“toolkit compartments”): agronomic factors, manufacturing recipe, food processing method and final preparation. These “tools” can be used selectively by the OIA, depending on the specific needs of each, in order to reduce the level of acrylamide in their products (EFSA, 2007, Food Drink Europe, 2019) [85,86].

According to the Regulation 2017/2158/EC (European Comission, 2017), the operators in the food industry are obliged to apply measures to reduce the level of AA, so as to reach the lowest possible levels, below the reference levels established in this normative act [81].

The mitigation measures must be adapted according to the nature of the activity of the operators in the field and include measures both in the agricultural sector and during the technological process of preparation of bread and bakery products [87].

In 2009 Codex Alimentarius drew up a guide "Code of practice for the reduction of acrylamide in foods" -CAC/RCP 67-2009, which specifies measures to reduce AA levels. Food business operators are advised to test in their own units all the measures recommended in this guide, in order to identify the optimal method to reduce this contaminant [88].

For the application of measures to reduce the level of AA in food, it must be taken into account that these methods do not affect the organoleptic and microbiological properties of finished products, their nutritional qualities and the associated consumer acceptability [88].

Worldwide there are three groups of strategies to reduce acrylamide formation: (i) modification of raw materials, (ii) optimization of processing conditions, and (iii) addition of exogenous additives [89].

If we compile all the mitigation measures recommended in the "toolkit" of EuropeDrinkFood, in the Guide developed by Codex Alimentarius, as well as those specified in the Regulation 2017/2158/EC (European Commission, 2017) [81], we find that they must be applied starting with the cereals cultivation stage and continuing throughout the technological process until the elaboration of baking instructions in the case of products intended to be completed at home or in public catering units. It has been established that AA formation in different bread types is influenced by the presence of free asparagine and reducing sugars that are associated with cultivar and crop’s type storage condition and harvest season [90].

### 4.1. Strategies to Reduce Acrylamide Formation in Bakery Products

In the agricultural sector, the aim will be to ensure good agricultural practices through fertilization (e.g., balanced level of sulfur in the soil, correct application of nitrogen) and good phytosanitary practices, so as to prevent the occurrence of fungal infections in cereals [73].

Because sulfur-poor soils increase the level of asparagine in cereals, and a large amount of sulfur causes the formation of sulfurous flours, which alter the organoleptic characteristics of finished products, it is recommended that sulfur fertilization be done in balanced doses. Excessive and delayed application of nitrogen fertilizers should be avoided so as not to create an environment conducive to increasing the amount of asparagine in cereals.

At the technological level, food industry operators can apply the following methods to reduce the AA level from bakery products shown in Figure 2.

In the case of food products to be completed after their purchase, clear technical specifications will be developed for the baking operation if this technological process is carried out at home or in public catering units.

The OIA is obliged to take and analyze samples from the products obtained, based on its own self-monitoring program, in order to monitor the level of acrylamide and to verify whether the mitigation measures are effective.

Internationally, several methods have been developed and implemented to reduce the level of AA in bakery products based in especially on the reduction of asparagine. Some examples of such methods are: adding enzymes such as asparaginase, acids and lacto-fermentation, the addition of polyvalent cations, addition of antioxidants, replacement or reduction of ammonium bicarbonate, such as ammonium salts in the case of finished bakery products, the optimization of the baking process, etc. Further, some of the best solutions and the most used ones reported so far to reduce asparagine in order to decrease AA level from bakery products, are presented in a detailed way.

### 4.2. Use of Asparaginase in Bakery Products

Nowadays, asparaginase is widely used to reduce AA levels in food. Asparaginase has the role of hydrolyzing asparagine to aspartic acid and ammonia before the heat treatment is applied and the Maillard reaction occurs. Asparginase is specific to asparagine and does not affect any other amino acids. Asparaginases from various sources including plants, bacteria, yeast and fungi, actinomycetes, algae and serum of some rodents have been isolated and characterized [91,92,93]. Numerous microorganisms may be valuable sources of asparaginase such as *Aspergillus tamarii*, *Photobacterium* spp., *Aerobacter* spp., *Bacillus* spp., *Xanthomonas* spp., *Serratia* spp., *Proteus vulgaris*, *Pseudomonas aeruginosa*, *Vibrio succinogenes* and *Streptomyces griseus*, etc. For therapeutic purposes, asparaginase is purified especially from *E. coli* or *Dickeya dadantii* whereas as an enzyme used for addition to foodstuffs the main sources of asparaginase are *Aspergillus oryzae* and *Aspergillus niger* [94,95]. In June 2007, JECFA analyzed asparaginase from *Aspergillus oryzae* and concluded from the available data that this enzyme does not induce a toxic effect on human health, but regulations were needed for its application nationally and internationally (JECFA, 2007). World Health Organization (WHO) included monographs on asparaginase from *Aspergillus oryzae* and *Aspergillus niger* in the 59th series of food additives (2008) [96] and 60th series (2009), respectively [97]. However, the asparaginase must be stable throughout the food processing and must not cause allergic or toxic reactions [98]. Nowadays, are available commercially for the market two products for their use in foodstuffs: PreventAse™ (DSM Food Specialties, Heerlen, the Netherlands) and Acrylaway^®^ (Novozymes, Bagsværd, Denmark) synthesized from *Aspergillus niger* and *Aspergillus oryzae*, respectively [99]. In the future it is possible to be available on the market asparginase from more microbial sources since FDA has suggested that is safe for food processing and a recombinant enzyme from *B. subtilis* source [100]. Currently, asparaginase is used in different foodstuffs in several countries, including the United States, Australia, New Zealand, China, Russia, Mexico, and several European countries [101].

The addition in the technological process of bakery products of asparaginase, aims to reduce the level of AA in the finished products [102]. Different studies have been reported that free asparagine in wheat flour was the main responsible for the acrylamide formation in bakery products in especially in biscuits which contains reduced sugars in their recipe and that way asparagine can reduce the acrylamide formation in bakery products [103,104]. Following laboratory tests performed on several foods, it was found that there is a significant decrease in AA levels, between 34–94%, for finished products to which the asparginase was added, compared to products to which this enzyme has not been added. The activity of asparaginase is influenced by the dose used, the time, temperature and pH to which the reaction takes place. Reducing AA by using asparaginase does not alter the sensorial and textural characteristics of the finished bakery products [105,106,107,108,109]. The effective dose for asparaginase was reported to be 200–1000 μg Kg^−1^ dough [110]. By using L-asparaginase produced by *Aspergillus oryzae* in savory and sweet biscuits a reduction between 34–92% was obtained [111,112]. It was concluded that this reduction depends on the biscuit recipe, i.e., water content, fat type, etc. The reduction of acrylamide was higher for biscuits with a low fat content and a higher amount of water [113]. Regarding the dose of asparaginase that must be used it was reported that a level of 500 U/Kg was enough for good results regarding the AA reduction level in biscuit products [30]. In gingerbread the reduction of AA was higher than 97% without any significant changes of the sensory characteristics of the finished product obtained from dough which was left for 48 h at ambient temperature with an enzyme addition level of 1000 U/kg of dough [114].

Also the addition different types of L-asparaginase from *Rhizomucor miehei* expressed in *Escherichia coli* to a level of 10 U/mg flour reduced the level of AA in biscuits by up to 80% [28]. The use of L-asparaginase produced from *Cladosporium* sp.at a level of 300 U/Kg reduced the AA content up to 73–97% in bread without any changes to its physical and sensorial characteristics [115]. From the rheological point of view no significant changes have been reported. However, a slight increase of water absorption value, dough stability, and a decrease of peak viscosity of wheat flour with increased level of asparaginase addition (50, 100, 200 and 300 U/kg) in wheat flour has been noticed.

### 4.3. Use of Polyvalent Cations in Bakery Products

Different studies have reported that different polyvalent cations may reduce the formation of acrylamide in bakery products [26]. Thus, different mono- and divalent cations such as Mg^2+^, Ca^2+^, K^+^ and Na^+^ may reduce the acrylamide formation during the Maillard reaction. This may be due to the fact that ionic associations between the charged groups on asparagine or related intermediates and ions added in wheat flour are involved. AA was reduced up to 74% and 52%, respectively, by magnesium and calcium addition in a chloride form to tortilla chips. In bakery products both chloride salts in the calcium and magnesium form affect dough rheological properties by reducing the development time and increasing the water absorption capacity [116,117]. In an asparagine -fructose system Ca^2+^ ions prevent AA formation completely, whereas the Na^+^ almost halved the AA formation. Also, its seems that Mg^2+^ and K^+^ have a similar effect to those of Ca^2+^ and Na^+^ cations, too [42]. In an asparagine–glucose model system the addition of sodium glutamate microcapsules reduced the AA level from 60.35% to 6.75% [21]. In a study on the use of different inorganic salts such as monovalent and divalent chlorides, hydrogen carbonates, phosphates and lactate in cereal matrices it was concluded that calcium chloride was the most efficient one, with a 90% acrylamide reduction in cereal systems. Sodium in the form of acid pyrophosphate and dihydrogen phosphate as well as potassium also decreased AA in cereal matrices by up to 75%, whereas calcium lactate, sodium chloride and potassium chloride decreased the AA level by up to 40–45%. Potassium and sodium hydrogen carbonates reduce the AA level only up to 30% in cereal matrices [118]. In a dough system has been reported that sodium and calcium in chloride form reduce the AA content by 23% and 36%, respectively [119]. In a cracker system heated to 180 °C for 15 min the addition of 1.3% NaCl also decreased the AA level [120]. However it seems that in the case of sodium this reduction is not constant. Claus et al. [121] have reported that with the addition of 1–2% sodium chloride the AA level decreased whereas at levels higher than 2% the AA level increased in bread and roll production at 220 °C for 30 min. Similar data has been reported also by Gökmen et al. [122] who concluded that AA increased in response to 5–20 µmol/L NaCl and decreased with 0.5–5 µmol/L NaCl addition in an asparagine–glucose model system. These data are favorable ones for bakery products since in general, in bread the salt (NaCl) addition level in wheat flour varies between 1.2–1.8%. In bread making salt not only has an important effect on the sensory properties of the baked products but also on the technological properties. Regarding the dough rheological properties salt addition creates a stronger gluten network increasing the dough development time, dough stability, dough extensibility and decreases water absorption capacity and degree of dough softening [123]. Concerning enzymatic activity, NaCl decreases the amylolytic and proteolytic activity. Regarding its effect on fermentation process, it seems that levels usually used in bread inhibit yeast activity, decreasing the gas formed during the fermentation process [124]. In bakery products NaCl leads to good sensory properties, increases the shelf life of the products, improves the loaf volume of bread and leads to a uniform crumb texture and brown colour [125,126].

Recently, more attention has been paid to calcium salts in order to prevent the appearance of AA in food, especially due to the fact that fortification with calcium salts of the wheat flour used in bakery products is a common practice [127]. However, it seems that calcium salt addition in bakery products affects the dough behavior during bread making in a significant way. Different studies have reported that the use of calcium in the gluconate and lactate form led to a strengthening effect on dough rheological properties, a decrease of dough softening, an increase in dough stability, an increase of gas formed during the fermentation process and amylase activity [128]. The use of calcium chloride also led to a strengthening effect on wheat flour dough and a decrease of the water absorption capacity [129]. Their use in the form of calcium carbonate or calcium chloride seems to reduce the AA levels in bakery products. However, the use of other calcium forms such as calcium propionate did not show a significant effect on AA reduction in the final bakery products [130]. Other studies have shown that the addition of 1% calcium chloride to wheat flour reduces the AA level in finished bakery products by 35% during baking [120]. Among different calcium salts (calcium lactate, calcium citrate, calcium carbonate and calcium acetate) added to wheat flour to a level of 0.1, 0.5 and 1% in cookie products, calcium carbonate was the most effective in reducing AA formation by up to 60% without any negative effects on the sensory scores. Good results have also been noticed for calcium citrate addition, whereas calcium lactate and calcium acetate have been described as less efficient for AA reduction [131]. However, Acar et al. [132] have reported that in cookies calcium lactate and calcium chloride added to dough up to a 0.5% level reduce the AA level by up to 70% without any significant changes on the sensory scores of the cookies. They also concluded that baking conditions may influence the calcium effect on AA reduction in bakery products. For baking temperatures between 150 and 250 °C the calcium chloride and lactate salts reduce the AA level in a significant way in cookie products.

### 4.4. Use of Acids and Lacto-Fermentation

Studies have shown that the formation of acrylamide depends on the pH value. The lower the pH is, the lower the AA levels in bakery products are. It seems that the addition of citric, lactic or ascorbic acid significantly reduces the level of AA in bakery products and is a relatively simple method to use [85,133]. The addition of tartaric and lactic acids, in 0.5% of bread recipes and to the biscuits and crispy bakery products, led to a linear reduction in AA levels [60,120]. Studies have been shown that by citric acid addition to the dough of semi-sweet biscuits, the pH is reduced and thus the concentration of AA in the finished products decreases by 20–30%. The addition of organic acids combined with pH adjustment can lead to low AA levels, but the addition of acids can also cause changes in sensorial characteristic of the finished product (less browning and modified product taste) [85].

The use of microorganisms such as lactic acid bacteria (LAB) and probiotics has been taken into consideration and can be successfully used to reduce the acrylamide content in bread [134]. This reduction is more related to the resulting pH decrease than to any consumption of precursor nutrients such as reducing sugars and asparagine by the microorganisms [21]. This reduction may be up to 75% when in the fermentation phase of bread making LAB are used [116]. The most commonly used genera of LAB for AA reduction are *Lactobacillus* and *Pediococcus* [134]. Some *Lactobacillus* species possess asparaginase genes that are yet to be characterized [135]. Some researchers have cloned the asparaginase gene from *L. reuteri* DSM20016 and *L. casei subsp. casei* ATCC 393 which were then expressed in *E. coli* [136,137]. Fermentation processes by LAB have the potential to be used for bread bio-preservation because they are safe for consumers [138,139,140,141]. These LAB naturally dominate in sourdough, and they produce metabolites that can inhibit fungal growth, and its consequences, including acrylamide content. For example, Dastmalchi et al. evaluated the action of LAB on reducing acrylamide content in bulk wheat bread. The AA levels in sourdough bread, especially bread fermented by *Lactobacillus paracasei* was lower than in the control sample by up to 45% [142]. Therefore, adding probiotics could represent a good promising approach in reducing the acrylamide levels in food products [90,143,144,145]. Another study showed that mitigation of acrylamide formation in bread can be achieved by reducing damaged starch in flour and by fermentation of the dough [146]. The use of LAB strains like *Lactobacillus plantarum*, *Lactobacillus brevis*, *Pediococcus acidilactici* and *Pediococcus pentoseus* in sourdough bread recipes reduces acrylamide formation by up to 84.7% (the best results being obtained with a *Pediococcus acidilactici* strain). The bread texture and flavor were also improved by its addition [1]. Another study reported that by combining *Aspergillus niger* glucoamylase and LAB strains AA formation may also be reduced in a significant way [29]. However, even if the use of sourdough in bread reduces the AA level its technological effect on bread making must be taken into account. Different studies have reported that sourdough addition in bread recipes has a weakning effect on wheat flour dough [124,126,147]. This may not favour the quality of bakery products obtained from weak quality wheat flour used bread making [148,149]. During mixing and extension sourdough addition causes an increase in the degree of softening, and a decrease of stability and dough extension. During pasting, sourdough addition causes an increase of the peak viscosity whereas during fermentation it causes an increase of the total volume of CO_2_ produced and the height under constraint of dough at maximum development time [124,126,147].

### 4.5. Use of Antioxidants in Bakery Products

A possible coupling with oxidative processes in bakery products led to several studies regarding the effects of antioxidants on acrylamide formation in baked products [23]. Different studies have reported on the effects of antioxidants such as ascorbyl palmitate, sodium ascorbate, butylated hydroxytoluene (BHT), vitamin E, sesamol, antioxidants from extracts of various spices and herbs such as oregano, cranberry, rosemary, cumin, anise, green tea, turmeric, ginger, etc. on acrylamide reduction in foodstuffs [2,23,150,151,152,153]. Nowadays there is an increasing trend towards the use of natural antioxidants obtained from spices and herb extracts in food products, including bakery ones, to reduce the formation of AA in these foods due to the fact that the consumers are demanding natural foods without preservatives. Moreover, it seems that natural antioxidants have a high effect on AA reduction in foods, larger than those obtained by using various additives as antioxidants. For example, it was reported that using ascorbyl palmitate and sodium ascorbate in a potato model did not reduce in a significant way its AA levels [154]. The use of BHT in lean meat led to an increase of its AA levels [23]. The use of natural antioxidants like those obtained from bamboo leaves and green tea extract significantly reduced the level of AA in toast. The Maillard reaction is prevented due to the ability of the green tea flavonoids such as epicatechin and epigallacatechin to capture carbonyl groups. The addition of antioxidants also inhibits the blocking of acrolein to a certain level, which can lead to lower AA levels. The effects of polyphenols on the Maillard reaction have been given particular attention by various researchers [155,156,157,158,159]. The use of different extracts obtained from buckwheat seeds and sprouts reduced the AA level in bread by 16.7–27.3% [160].

The addition of spices such as rosemary to wheat flour helps reduce the formation of AA when it is added to the dough before the heat treatment is applied [2]. Antioxidants such as phenolic compounds, flavonoids, vitamins, and phenolic extracts from various spices (thyme, cumin and anise and so on), herb extracts (green tea, mint, fennel, turmeric) have been reported as inhibiting acrylamide formation [150,151,152]. One of the most used natural antioxidants in foodstuffs, including bakery ones, in order to reduce the AA content is rosemary (*Rosmarinus officinalis* L.), for which very good results have been reported. In commercial potato chips rosemary reduces the AA levels by up to 25% [161], while the use of oregano in fried potato slices did not show any significant effect [162]. The use of rosemary extract in shortcrust cookies to a level of 0.1–0.5% did not affect in a significant way the cookies’ sensory properties and reduced the AA content, especially when 0.5% rosemary extract was added to wheat flour [2]. The use of two aqueous extracts from two rosemary species at a level of 1 and 10% in wheat buns decreased the AA formation by up to 67%, while extracts from dittany (*Origanus dictamnus* L.) slightly increased the AA formation. It was concluded that an increase of the aqueous rosemary extract level to 10% did not decrease the acrylamide content further compared to the addition of a 1% extract [23].

### 4.6. Total or Partial Replacement of Ammonium Bicarbonate in Bakery Products

Ammonium bicarbonate is one of the critical factors in AA formation in bakery products by acting as an additional source of nitrogen or by indirectly catalyzing the conversion of sugars to produce reactive carbonyls [163]. However, its influence on AA levels in baked products is low if reducing sugars are present. Also, its promoting effect is not related to free asparagine in the raw materials used in bakery recipes [164]. Replacement of ammonium bicarbonate with alternative leavening agents such as baking soda and acidulates, sodium bicarbonate and disodium diphosphates with organic acids (tartaric acid, citric acid) or their potassium variants like potassium bicarbonate with sodium acid pyophosphate, may be an important strategy to reduce the AA levels in bakery products [164,165,166]. Different studies have been reported that the replacement of ammonium carbonate or bicarbonate with the corresponding sodium salt may cause an acrylamide reduction of up to 70% [164,166]. Replacing ammonium bicarbonate with baking soda reduced the formation of AA levels in bakery products. Studies on biscuits and ginger bread in which ammonium bicarbonate was replaced by sodium bicarbonate reported a significant reduction of AA levels in these products [85].

### 4.7. Processing Methods

The thermal energy is one of the most important aspects that must be taken into account in AA formation in bakery products. Lowering the temperature during baking decreases the AA formation in baking products, but this requires longer baking times in order to achieve the desired final moisture content. For example, breads baked at 200 °C/70 min presented an AA level of 124.1 mg/kg, whereas bread baked to 240 °C/50 min presented an AA level of 92.4 mg/kg without any significant changes regarding the sensory characteristics including flavor and colour [121]. In order to decrease the baking time and temperature value some modern process may be used in combination with conventional baking or not. One of these is the use of a radio frequency (RF) heating method to reduce the AA level is bakery products since RF generates heat inside the baked products where most of the moisture remains after baking. A decrease of AA formation in leavened cake and short dough biscuits by the use of RF has been reported by several studies [167,168]. Combining the RF post-drying process (45 s) and conventional baking (205 °C for 8 min) the AA formation in biscuits may be reduced by up to 50% [168]. Microwave heating can also be a method to reduce AA formation in bakery products. However, during the baking process the use of microwaves may led to products with weak sensory characteristics with insufficient brown color on the surface and poor crust formation of breads [31]. The combination of microwaves and the classical baking process is recommended to reduce AA content and to increase the quality of bakery products [169,170]. In biscuit products microwave treatment (700 W/90 s) reduces the AA level by 10% compared to those obtained through a traditional method (190 °C/10 min) [158]. Baking under nitrogen or carbon dioxide atmospheres led to a 50% reduction in bread whereas baking in a sulphur dioxide atmosphere led to a 99% reduction of AA in bread [32]. Steam and falling temperature baking decreased the AA level in bread products giving products with an acceptable crust colour. Moreover the lowest AA levels were produced especially by steam baking. The use of steam for at least 10 minutes of baking reduced the AA level almost 50% [171]. An extrusion cooking method with CO_2_ injection decreased the AA level of extrudates up to 82% and increased the feed moisture content from 22 to 24% [172]. The use of microencapsulation may be a promising technique to be used in bakery products. For example the use of sodium glutamate microcapsules in an asparagine–glucose system may reduce AA formation by up to 60.35% [21]. The use of partial vacuum in the last phase of baking may be an effective strategy to reduce the AA level in cookies by up to 53% due to the fact that the desired moisture content was obtained in a shorter period of time when vacuum was applied [173].

## 5. Perspectives

Given the risks of AA to human health, food producers and industry authorities need to continue the process of reducing AA levels from foods. The complete elimination of AA from foods is not possible, so food producers are trying to reduce this contaminant to the lowest levels possible. However, it seems that the classical methods used to reduce the AA formation in foods such as changes in product composition and/or technological process conditions may have undesirable effects on nutritional, sensory characteristics (change in taste, texture, color, etc.) and food safety (inadequate reduction of microbial population, natural toxins, etc.).

In the future, different methods may be used to reduce the AA level from foods, including naturally ones (such as antioxidants) that may keep the labels of bakery products clean. Consumers and bakery producers only want to use methods that avoid compromising the chemical and microbiological safety of food. It can be noticed that in certain situations, that there is a risk of accidental formation of other undesirable substances and process contaminants, such as 3-monochlorpropane-1,2-diol (3-MCPD), aldehydes, hydroxymethylfurfural, N-nitrosamines, polycyclic aromatic hydrocarbons, amines, furans, heterocyclic aromatic amines and aminoacid pyrolysis products, etc. There is also a potential risk of changing the baking process parameters with negative effect on foods safety and bakery sensory characteristics. That way nowadays different modern methods of processing are being tested, especially during the baking phase (radiofrequency, inert atmospheres, steam baking, microwave baking, partial vacuum, etc.) in order to reduce the AA levels without significant changes to the quality of the baking products.

It is well known that AA formation is closely associated with some sensory characteristics, especially the color and aroma produced in the case of bakery products after baking. Therefore, any method of reducing the formation and the level of AA must be evaluated from the perspective of the acceptability of the final product by consumers.

To produce bakery products with a low level of acrylamide, with good sensory and technological characteristics further studies are recommended in the future to combine different types of raw materials with a low content of asparagine and reducing sugars, as well as the use of exogenous ingredients, to optimize bakery product recipes and processes, obtaining finished products as natural as possible, safe and well received by consumers.

Therefore, agricultural practices, manufacturing recipes and technological processes all need to be improved to reduce the AA levels in bakery products and to avoid their rejection by consumers. That is why it is important for bread making producers to carry out a careful evaluation of the reduction methods applied so as to take into account the real composition of their products, the production equipment they have and the need to provide consumers with quality products, in line with their brand image and consumer expectations. In the future, it is expected that there will be increased number of scientific studies on acrylamide reduction in bakery products while maintaining the bakery products’ quality characteristics demanded by the consumers.

## 6. Conclusions

The presence of acrylamide in bakery products is nowadays one of the most difficult problems facing the bread-making industry. Furthermore, because bakery products are among the most consumed ones in the world, the levels of acrylamide in them, which are formed during the baking process lead to a unavoidable exposure to AA through bakery product intake. Therefore, it is necessary to know the toxic effects of AA on the human body and how to reduce its content in order to obtain safe products. For this purpose, the European Union has established different maximum levels of AA for every food type which must be monitored by the food safety authorities. The compliance with the maximum benchmark levels of AA set by the European Union is extremely difficult for the bakery industry and it forcing producers to engage on the acrylamide issue in order to maintain the AA levels in bakery products within European regulations.

Reducing the AA level in bakery products may be achieved by changing the parameters of the technological process (i.e., baking at a lower temperature) or by using new alternative baking techniques, by using raw materials with low asparagine and reducing sugars content or by using some ingredients (different acids which lower the pH, divalent ions, natural antioxidants, asparagine from different sources, sourdough-producing LAB, etc.). Due to the fact that AA induces cancer more actions must be taken both by food producers and legislative authorities to reduce its levels in foodstuffs. Taking into account the toxic effect of AA on the human body lowering its level in the most consumed foodstuffs in the world, bakery products, should be a priority in risk management for bakery producers.

## Figures and Tables

**Figure 1 ijerph-18-04332-f001:**
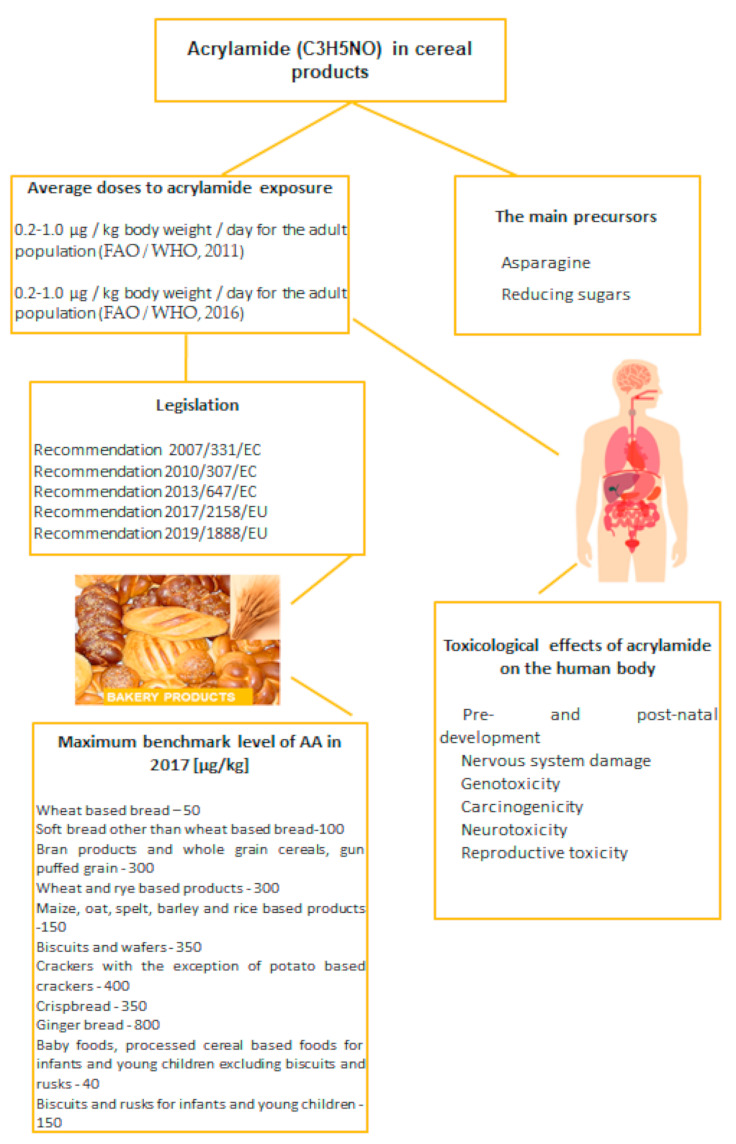
Overview of acrylamide, its toxicity and legal regulations in bakery industry.

**Figure 2 ijerph-18-04332-f002:**
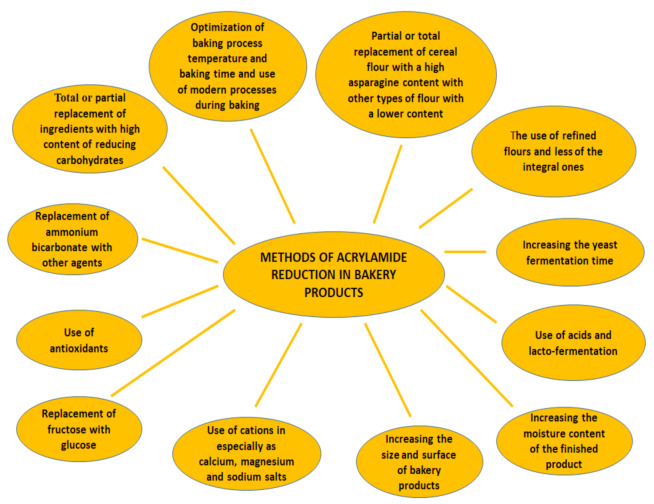
Methods of acrylamide reduction in bakery products.

**Table 1 ijerph-18-04332-t001:** Indicative values for acrylamide in bakery products set by the European Commission for 2007–2021 periods (2013/647/EU; 2017/2158/EU).

Food	Benchmark Level 2013 [μg/kg]	Benchmark Level 2017 [μg/kg]
Soft bread		
Wheat based bread	80	50
Soft bread other than wheat based bread	150	100
Breakfast cereals (excl. porridge)		
Bran products and whole grain cereals, gun puffed grain	400	300
Wheat and rye based products	300	300
Maize, oat, spelt, barley and rice based products	200	150
Biscuits and wafers	500	350
Crackers with the exception of potato based crackers	500	400
Crispbread	450	350
Ginger bread	1000	800
Baby foods, processed cereal based foods for infants and young children excluding biscuits and rusks	50	40
Biscuits and rusks for infants and young children	200	150

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
