# Peer review of "Acrylamide in Bakery Products: A Review on Health Risks, Legal Regulations and Strategies to Reduce Its Formation"

_ijerph, 2021, doi:10.3390/ijerph18084332_

Round 1

Reviewer 1 Report

Overview and general recommendation:

the Sarion et al.’s manuscript deal with acrylamide content in bakery products, health risks following it assumption, regulations and some strategies to reduce its formation during production. Despite the topic is well known the review is interesting because it focuses on a specific food category. I find it clear, well written with a wide bibliograph.

Nevertheless, there are several small errors in the text. I attached the pdf file with some corrections, but I suggest checking the entire document.

In my opinion the figure 1 is very useful to understand the scheme of the review but it should be more graphically cured.  

In order to improve reading the paragraphs could be divided into sub-paragraphs. For example: the methods of analysis describe from line 207 could be a subparagraph of the paragraph 3; or, in the paragraph 4 the description of the strategies to reduce acrylamide formation should be a separated subparagraph.

Author Response

8 April 2021

Dear Referee,  

We would like to thank the referee for the close reading and for the proper suggestions. We hope that we provide all the answers to the reviewer’s comments.

Thank you very much for the recommendations to publish our paper entitled “Acrylamide in bakery products: a review on health risks, legal regulations and strategies to reduce it formation”.

The present version of the paper has been revised according to the reviewer’s suggestions.             

GENERAL COMMENTS:

Referee comments: Overview and general recommendation:

the Sarion et al.’s manuscript deal with acrylamide content in bakery products, health risks following it assumption, regulations and some strategies to reduce its formation during production. Despite the topic is well known the review is interesting because it focuses on a specific food category. I find it clear, well written with a wide bibliograph.

Response: We would like to thank to the referee for its appreciations. We hope we made all the corrections that he/she requested in order to improve our manuscript.

Referee comments: Nevertheless, there are several small errors in the text. I attached the pdf file with some corrections, but I suggest checking the entire document.

Response: We would like to thank to the referee for the close reading of our manuscript. We revised the manuscript according to the corrections inserted by the referee into the pdf file of our manuscript. Also we checked the entire manuscript once again.

Referee comments: In my opinion the figure 1 is very useful to understand the scheme of the review but it should be more graphically cured.  

Response: We would like to thank to the referee for his/her appreciations. We improved figure 1 for the graphical point of view.

Referee comments: In order to improve reading the paragraphs could be divided into sub-paragraphs. For example: the methods of analysis describe from line 207 could be a subparagraph of the paragraph 3; or, in the paragraph 4 the description of the strategies to reduce acrylamide formation should be a separated subparagraph.

Response: We would like to thank to the referee for his/her suggestions. We revised the paragraph 3 by introducing some aspects related to the toxicological effects in paragraph 2 and now this paragraph underline in a better way the legislative aspects. It presents only legislative aspects and  that way, it is not possible to separated it in different subparagraphs. However, we agree with the referee point of view and we added another subparagraph in chapter 4 according to the referee suggestions.

Reviewer 2 Report

The manuscript presents a comprehensive review of acrylamide legislation, toxicity effects and measure to reduce its formation with a special focus on bakery products. 

The review is suitable for publication. However, the authors should improve the manuscript on some points:

  1. Paragraph 2 and 3 need re-arrangement as the two paragraphs overlap each other, repeating the same concepts or findings. For instance, I would suggest telling in chronological order how toxicological findings helped in developing the legislation. However, I leave the authors free to find other appropriate re-arrangement.
  2. Probably, a single paragraph on sampling and detection methods available to food control authority and food business operators would also be appropriate, discussing their LOD/LOQ. Are there any rapid methods? The sampling methods are discussed appropriately. However, some paragraphs need improvement in the English style (as an example, please refer to line 320-323)
  3. In paragraph 4.6, lines 632-638, the authors wrote, " Lowering temperature during baking decreases the AA formation in baking products", and then they offer an example where data suggests the contrary. Is this correct?
  4. English style needs some revision.

Author Response

8 April 2021

Dear Referee,  

We would like to thank the referee for the close reading and for the proper suggestions. We hope that we provide all the answers to the reviewer’s comments.

Thank you very much for the recommendations to publish our paper entitled “Acrylamide in bakery products: a review on health risks, legal regulations and strategies to reduce it formation”.

The present version of the paper has been revised according to the reviewer’s suggestions.             

GENERAL COMMENTS:

Referee comments: The manuscript presents a comprehensive review of acrylamide legislation, toxicity effects and measure to reduce its formation with a special focus on bakery products. 

The review is suitable for publication.

However, the authors should improve the manuscript on some points:

Referee comments: Paragraph 2 and 3 need re-arrangement as the two paragraphs overlap each other, repeating the same concepts or findings. For instance, I would suggest telling in chronological order how toxicological findings helped in developing the legislation. However, I leave the authors free to find other appropriate re-arrangement.

Response: We would like to thank to the referee for his/her observations. Yes, indeed the first paragraphs from chapter 3 are more related to chapter 2 subject. We arrange now the chapter 2 and chapter 3 once again according to referee suggestions. We arrange chapter 2 in chronological order, namely how toxicological findings helped in developing the legislation.

Referee comments: Probably, a single paragraph on sampling and detection methods available to food control authority and food business operators would also be appropriate, discussing their LOD/LOQ. Are there any rapid methods? The sampling methods are discussed appropriately. However, some paragraphs need improvement in the English style (as an example, please refer to line 320-323)

Response: We would like to thank to the referee for his/her observations. We completed now the manuscript with a paragraph on sampling and detection methods available to food control authority and food business operators would also be appropriate, discussing their LOD/LOQ. Also, we discussed about the possibility of a rapid method for AA determination. We improved the paragraph from English point of view (line 320-232) as referee suggested.

Referee comments: In paragraph 4.6, lines 632-638, the authors wrote, "Lowering temperature during baking decreases the AA formation in baking products", and then they offer an example where data suggests the contrary. Is this correct?

Response: We would like to thank to the referee for the close reading of our manuscript. Yes, the affirmations it is a correct one. It is true that an example is offer further that breads baked at 2000C/70 min presented an AA level of 124.1 mg/kg whereas the bread baked to 2400C/50 min presented an AA level of 92.4 mg/kg but the time of baking is one of the factors that influence the AA level too and therefore the results reported are not contradictory ones.

Referee comments: English style needs some revision.

Response: The article was once again revised from the English point of view.